# Elementary Teachers' Perceptions and Enactment of Supplemental, Game-Enhanced Fraction Intervention

Jessica Hunt [1,*], Michelle Taub [2], Alejandra Duarte [1], Brianna Bentley [1], Kelly Womack-Adams [1], Matthew Marino [2], Kenneth Holman [2] and Adrian Kuhlman [1]

1  College of Education, North Carolina State University, Raleigh, NC 27695, USA; aduarte2@ncsu.edu (A.D.); brbentle@ncsu.edu (B.B.); kawomack@ncsu.edu (K.W.-A.); akuhlma@ncsu.edu (A.K.)
2  College of Community Innovation and Education, University of Central Florida, Orlando, FL 32816, USA; michelle.taub@ucf.edu (M.T.); matthew.marino@ucf.edu (M.M.); kenneth.holman@ucf.edu (K.H.)
*  Correspondence: jhunt5@ncsu.edu

**Abstract:** Curricula enhanced through the use of digital games can benefit students in their interest and learning of Science, Technology, Engineering, and Mathematics (STEM) concepts. Elementary teachers' likelihood to embrace and use game-enhanced instructional approaches with integrity in mathematics has not been extensively studied. In this study, a sequential mixed methods design was employed to investigate the feasibility of a game-enhanced supplemental fraction curriculum in elementary classrooms, including how teachers implemented the curriculum, their perspectives and experiences as they used it, and their students' resulting fraction learning and STEM interest. Teachers implemented the supplemental curriculum with varying adherence but had common experiences throughout their implementation. Teachers expressed experiences related to (1) time, (2) curriculum being too different, and (3) too difficult for students. Their strategies to handle those phenomena varied. Teachers that demonstrated higher adherence to the game-enhanced supplemental fraction curriculum had students that displayed higher STEM interest and fraction learning. While this study helps to better understand elementary teachers' experiences with game-enhanced mathematics curricula, implications for further research and program development are also discussed.

**Keywords:** digital games; fractions; elementary mathematics; teacher experiences; mixed methods

## 1. Introduction

Science, Technology, Engineering, and Mathematics (STEM) fields as well as Information, Communication, and Technology (ICT) careers benefit from the expertise of a diverse workforce [1]. Issues of access and equity related to STEM and ICT fields often begin in elementary school, where students from historically excluded populations have limited access to mathematics instruction that links to STEM and ICT careers [2]. For example, typical instructional practices may provide limited ways in which students can engage, represent, or express their STEM knowledge; may not connect students' knowledge and/or experiences to STEM or ICT; or may not be engaging for students [3–5].

Game-enhanced mathematics curricula may be a way for elementary teachers to promote access, improve engagement, and empower their students by bolstering their learning outcomes and their interest in foundational STEM and ICT content [6–10]. Games can bolster students' problem-solving and situate students' learning opportunities in problem based scenarios through multiple representations, tools, or solution strategies that embrace students' prior knowledge [6,7]. They can also link foundational mathematics content to STEM and ICT careers, potentially increasing students' engagement and learning outcomes (e.g., [6]).

However, elementary teachers' propensity to embrace and use game-enhanced instructional approaches with integrity in mathematics is not well understood. Teachers ultimately make decisions regarding how and to what extent curricular experiences, such

as games, are implemented in classrooms [11]. While much research exists on middle- and high-school teachers' use of game-enhanced programs in mathematics, there is little research that focuses on elementary school teachers [12,13]. Therefore, it is necessary to understand how elementary school teachers use game-enhanced curricula with their students, what their successes and challenges are in implementation, and the extent to which students' outcomes might change as a result of different teacher implementations.

The purpose of this paper is to explore how elementary school teachers perceive and implement a game-enhanced supplemental curriculum for fractions called Model Mathematics Education (ModelME). ModelME is a 36-lesson supplemental curriculum with a game built into it. The program is designed to increase student engagement, fraction knowledge, and STEM/ICT career interest and is designed using the Universal Design for Learning (UDL) framework, an efficacious design framework for accessible and equitable instructional materials [3]. A sequential mixed methods design is employed to investigate the feasibility of the curriculum in elementary classrooms, including how teachers implemented the curriculum, their perspectives and experiences as they used it, and their students' resulting fraction learning and STEM interest. The research questions are:

1.  To what extent do elementary teachers implement a game-enhanced supplemental fraction curriculum with integrity?
2.  What are elementary teachers' experiences and perspectives after implementing the game-enhanced fraction intervention in their classrooms?
3.  To what extent did students' fraction schemes and STEM interest change after participating in a game-enhanced intervention?

## 2. Literature Review

### 2.1. Game-Enhanced Instruction in Elementary School

Over the past 20 years, digital game-based mathematics instruction has emerged as a means to improve student outcomes, such as motivation, engagement, and learning, for a diverse array of students [14–18]. Games can be important and helpful instructional tools in elementary mathematics classrooms because they can be used to: (1) supplement core instructional content, (2) serve as alternative or unique instructional experiences across different STEM content (e.g., [10,15,16]), and (3) provide additional instructional support (e.g., [17]). The potential of games to enhance STEM content accessibility, increase collaborative problem-solving, and allow for exploration of mathematics concepts in innovative ways is also promising [10,15]. For example, games have been developed to enhance content accessibility and increase problem solving as well as improve cognitive functioning [18–23].

The research base that examines the impact of game-based instruction on student outcomes in mathematics, specifically, is growing. For example, we know that digital game-based instruction can positively impact learning in mathematics [4], with larger effects typically found for students in middle- or high-school [7,24]. Smaller gains are documented in elementary school [12], although the overall use of games by elementary school students is historically under-researched [19,25]. Research also shows that the propensity of games to produce positive effects on student outcomes rests in how and to what extent teachers use games to support content learning as well as the training, development, and support they receive to do so [11]. In this study, we posited that the extent to which our game enhanced program aligned with teachers' experiences and beliefs regarding instruction would also impact the teachers' use of games in important ways.

### 2.2. Theoretical Framework: Teachers' Beliefs and Use of Game-Enhanced Interventions

Each individual has their own set of beliefs which strongly direct their perceptions within their particular context [26]. In the case of the classroom, a teacher's beliefs can include views about themselves, their roles and responsibilities, their students, their subject, and the curriculum material. Beliefs surrounding educational topics and issues can

influence teacher actions within their classroom, specifically whether or not practices or programs are implemented with fidelity in those classrooms. For example, when required to implement technology practices, teachers' beliefs about knowing and learning impact their implementation [11]. Moreover, because teacher beliefs impact the structure and climate of the classroom that students are a part of, teacher beliefs also impact students' learning [27].

One technology tool with increased prominence in the elementary mathematics classroom is digital games. Teachers' beliefs surrounding the use of digital games to bolster student learning is documented (e.g., [28,29]), with varied findings. One finding is that teachers prefer more traditional teaching methods (e.g., explanation of concepts, repeated drills, and practice) over the use of digital games for mathematics. Teachers were open to using digital games if they perceived that it could increase their students' mathematics performance [29]. Another finding is that teachers may link the use of digital games to increased engagement while learning new content as well as increased opportunities for practice to master concepts, something teachers see as necessary pedagogically [28]. At the same time, teachers may view digital games as a distraction and not as advantageous for student practice as other tools, such as physical manipulatives. While some teachers believe that the games can be useful, they are not always viewed as the most beneficial tools of instruction and thus not always worth the time the games would take to incorporate into their teaching [30,31]. A fourth finding rests in the additional teaching load that including digital games brings for teachers, even amidst overall positive views of the use of digital games in the mathematics classroom more generally [29].

The varied beliefs and attitudes teachers hold towards the use of digital games in the mathematics classroom play a role in if the games are used, and how. For instance, Yeo et al. [32] showed there was a direct relationship between teachers' attitudes surrounding games' use to the likelihood of the games actually being implemented as intended in the classroom. But, this relationship was also mediated by teachers' perceptions of other environmental factors, such as how well it fit into the curriculum or the time afforded them to use the games in class. Even with positively held beliefs towards digital games, teachers occasionally hesitate to implement them in the classroom due to factors such as cost and an educational focus on increasing scores on standardized tests [33]. So, while beliefs held by the teacher about using digital games in math could indicate the likelihood of implementing them [32], it is not always the case [30,32].

There is a small amount of research that illustrates elementary school teachers' perspectives on and experiences with the use and implementation of digital games in mathematics and the challenges teachers face. Time is a consistent theme: concern over lack of time to be able to successfully implement digital games to support mathematics learning was a common concern (e.g., [34,35]). For example, in a survey conducted in 2002 and then again in 2003, time was listed as a major obstacle in effectively utilizing games in the classroom [34]. Teachers stated they do not have enough time to implement games that they believe are not directly instructing students and that they themselves have not had sufficient time to become familiar with those games. However, when teachers are able to have more experience with a digital game, they are much more confident in its effectiveness and how to best support their students [36].

### 2.3. The Current Study

The current study investigates how elementary teachers implement a game-enhanced supplemental fraction curriculum, their experiences and perspectives, and the extent to which students' fraction schemes and STEM interest change after participating in a game-enhanced intervention. The game-enhanced curriculum investigated in this study is a supplemental program that sits on top of students' core mathematics curriculum [6,37,38]. Teachers use the program with their students three days a week for 35 min a day. Core components of the program include multiple means of expression (MME), representation

(MMR), and engagement (MME), carefully selected tasks, cognitive prompts, and social mediation of learning.

First, MME, MMR, and MME are supported by a choice of problem-solving methods, materials, and tools [4] that players can select based on their preferences in the game's user interface. Additionally, the sandbox play design of the game allows students to create fractions using various actions, such as iterating, partitioning, or splitting quantities, in a way that encourages risk taking and yields opportunities to learn from mistakes. Second, the task challenges that students encounter in the game are based on documented pathways that students use to generate and, eventually, abstract fractional quantities [6,37,38]. Third, gameplay is supported by the game's cognitive prompts. Cognitive prompts are designed to support the students' generation of goals, monitoring of problem solving, and reflection on the results of their game actions—techniques that are largely underutilized in math instruction [4,37–40]. Finally, the curriculum integrates students' gameplay with opportunities to engage in mathematical explanation, and justification through one of three pedagogical routines: a worked example, a game replay, or a number string.

To prepare teachers to implement the supplemental curriculum, researchers provided four one-half day training sessions. Day one opened with the study's purpose, the target population, and the theory of change and logic model for the overall project. For the next two days, teachers studied student gameplay to deepen their understanding of how the core program components are used to bolster student learning and played the game themselves as learners. Researchers gave teachers a curriculum guide on the final day to drive small group practice opportunities, where teachers delivered the curriculum using the curriculum guide as a resource through role playing, rotating between teaching roles and student roles. For each role, the groups also engaged in the after-game tasks, discourse, and talk moves to facilitate a sample student conversation. Finally, researchers prepared the teachers to administer the study's measures.

## 3. Methods

### 3.1. Participants and Setting

Six fourth- and fifth-grade teachers and their students ($n = 133$) in two different schools in the southeast United States participated in the study. Each school was located in a rural setting and included students with intersecting identities in terms of race, language, and disability. The supplemental curriculum was administered by the teachers in their core mathematics classrooms, which included approximately 15–25 students with each teacher. The program took place over nine weeks, which is considered best practice in terms of time period for technology-based interventions [41]. Prior to the study, informed consent and assent were gathered from teacher and student participants using Institutional Review Board (IRB) approved documents. Demographic information for the six participating teachers and their students is given in Table 1.

**Table 1.** Teacher and student demographics.

| Teachers | | | | | Students | | | | |
|---|---|---|---|---|---|---|---|---|---|
| School | Gender | Yrs. Exp | Math Content Hours | Math Ped Hours | School | Grade | Gender | Race | Dis. Status |
| 1 (67%) | Female (100%) | 2–5 (16%) | <6 (0%) | <6 (17%) | 1 (44%) | 4th (43%) | Female (41%) | Hispanic (28%) | Yes (16%) |
| 2 (33%) | Male (0%) | 6–20 (68%) | 6–12 (50%) | 6–12 (50%) | 2 (56%) | 5th (57%) | Male (59%) | White (32%) | No (84%) |
| | | >20 (16%) | >12 (50%) | >12 (33%) | | | | African American (21%) | |
| | | | | | | | | 2 or more races (19%) | |

### 3.2. Data Sources and Measures

To understand the extent to which teachers implemented the program with integrity, we observed approximately 35% of all teachers' lesson enactments. Teachers audiotaped

all lessons; lessons were selected for review via a random number generator. Once a lesson was identified for review by the random generator, that lesson was observed for all teachers using a checklist (see Appendix A and data collection procedures), provided the tape was of acceptable quality to listen to the lesson in its entirety. If it was not, then a lesson immediately preceding or following the selected lesson was observed.

Teacher focus groups were held at the conclusion of the study to gauge teacher perspectives on, and overall experiences implementing, the game-enhanced curriculum. Two separate focus groups were held and distinguished by school. The focus groups were held virtually using semi-structured questions and lasted anywhere from 45 to 60 min. Participants were not compensated for their time. Focus-group questions included asking teachers about the kinds of student data they would like to access in the game, the strengths and weaknesses of the curriculum and its materials (e.g., embedded game, previews, after game tasks), aspects of the curriculum teachers found easy or difficult to implement in their classrooms, their perspectives regarding the amount of scripting provided, whether or not they would change anything about the curriculum and, if so, what they would change and why, and aspects of the embedded game in particular that the teachers saw as beneficial or challenging. See Appendix A for the full list of questions and protocol.

Finally, to gauge changes in students' fraction schemes and STEM interest, two measures were used: (a) the Fraction Schemes Test [42] and (b) the Upper Elementary School (4–5) Student Attitudes Toward STEM (S-STEM) survey [43]. The Fraction Schemes test is a 12-item measure of the effects of the intervention on students' fraction conceptions. Internal consistency reliability for the test was reported as 0.70; criterion-related validity was reported as 0.58 ($p < 0.01$). The S-STEM was developed as part of a US National Science Foundation (NSF) funded research program and measures students' confidence and self-efficacy in STEM subjects, 21st century learning skills, and interests in STEM careers. It contains 56 items across six constructs. For this study, the math attitudes (8 items) construct and the interest in STEM career areas construct were used (12 items). Responses are supported by a five-point Likert scale, with response options ranging from "strongly disagree" (1) to "strongly agree" (5). Higher scores reflect the greater perceived value of participants. Cronbach's $\alpha$ of the S-STEM ranged from 0.84 to 0.86 for the grade 4–5 subscales.

### 3.3. Procedures

3.3.1. Intervention Procedures

Teachers followed the curriculum guide to teach the supplemental program for nine consecutive weeks, 35 min a day, three days a week. Each lesson contained a five-minute preview, 10–15 min of student gameplay, and a 15–20 min after-game task (i.e., a number string or a game replay). Previews were supported by videos and were often presented with questions for students to discuss. One preview showed students the location and prevalence of windmills across the country as well as in the students' state and counties, while others illustrated how wind turbines work. Students were invited during the previews to elaborate, converse, or ask questions about a STEM or ICT career depicted in the game. After the preview, students played the universally designed video game for 10–15 min. The game presents fraction challenges along a learning trajectory that spans five game worlds using sandbox, puzzle-like play. Students play the role of "Bunny", a character whose attributes they can change according to their preferences.

Fraction challenges range in concepts from unit fractions to partitive (i.e., non-unit) fractions, to reversible fractions (e.g., using a non-unit fraction to produce a whole), and to multiplicative concepts (e.g., taking a part of a part, distributing $m$ fractional units over $n$ whole units) [37,44]. Some challenges are more constrained to invite particular actions on objects (e.g., find the length of exactly one share of a rectangular length) while other challenges invite a range of approaches and ways of quantification. In every challenge, players are given a choice of tools that they can use to engage with the fraction challenges. The current version of the game is leveled; that is, students must successfully complete each

challenge in order to advance to the next challenge, subworld, and game world. Gameplay is individualized and saves student progress along the way. As students play, different features of the character customizer "unlock".

After gameplay, students engage in a 15–20 min discussion with their classmates and their teacher through number strings and either a game replay or a worked example. Game replay procedures resemble think-pair-shares where students are given a game world challenge to recreate their strategy (four minutes). Students then pair with another classmate and share their strategies (four to five minutes). To do so, they are given a structure called 'talk and share', where sentence stems provide support for students to both share (e.g., "my strategy was. . ."; "my strategy makes sense to me because. . .") and listen to and question what is being shared (e.g., "I noticed that you said. . ."; "I had a question about. . ."). Students wrap up the game replay in a whole group share out, where they share their strategies around the discussion focus for the lesson (e.g., partitioning or iterating strategy to make n/n from m/n) and the overall learning goal for the world (e.g., m/n reflects a multiplicative relationship between the denominator—how many times 1/n repeats to form n/n—and the numerator—how many 1/n are being considered). Procedures for worked examples are similar, with the exception that teachers show students' worked solutions for problems they encountered in the game that were either correct, incorrect, or partially correct. Thus, students spend the think, pair, and share writing about their analysis of the worked examples by responding to given thinking prompts (e.g., "What did. . . do first?"; "Do you think . . . would have gotten a different answer if they. . .?").

Procedures for number strings involved students being shown a problem and asked to solve it in their heads and indicate with a nonverbal symbol when they had a solution. The teacher then called on students to share their reasoning; teachers used a core representation (e.g., a number line) to illustrate students' strategies. The process was then repeated with two to four additional problems; with each repetition, teachers asked students if they could use their thinking from the previous problems to engage with the newly presented problem.

### 3.3.2. Data Collection Procedures

Qualitative data collection. One week after the conclusion of the final curriculum lessons, teachers were gathered into focus groups to inquire about their experiences and perspectives implementing the game-enhanced curriculum. There were four teachers in one focus group and two teachers in another (i.e., one focus group per school site). Teachers were sent a Zoom link to take part in each focus group. Upon logging in, the lead facilitators read a standard statement about the purpose of the focus group and that teachers were free to participate as little or as much as they felt comfortable. Teachers were asked if they gave permission for the Zoom session to be recorded; upon consent, the facilitator began the focus group and asked each question one at a time. A secondary facilitator took detailed notes of the discussion, prompted the lead facilitator on time, and made a notation each time a participant made a comment to note trends. At times, the facilitators would restate teachers' responses for member checking and would also ask questions to follow up or gain elaboration on a response.

Quantitative data collection. To gauge teachers' integrity to the curriculum, a research assistant listened to each selected lesson tape while completing a checklist (see Appendix A) to identify the essential points the teacher implemented. The checklists are a point-by-point accounting of each statement and teacher action required by the curriculum. The research assistant checked off each item as it was enacted by the teacher as captured by the recording. The research assistant also recorded, at the top of each checklist, the timing of each lesson (i.e., the amount of time the teacher spent in each lesson segment). They also added descriptive information into sections embedded into the checklist areas of the curriculum that teachers added to, deleted, or changed.

Teachers were also asked to administer the Test of Fraction Schemes and the S-STEM survey to their students in their whole classroom settings in the morning hours right

after the start of the school day. Students completed these assessments right before and immediately following the nine-week curriculum program. Students were given 30 min each to complete the pre- and the post-test. Teachers read specific directions to students (i.e., told students to write their name at the top of the tests, and to do their best work) and gave no other direction, except to read aloud questions upon individual student requests. Similar procedures were used for the surveys, with a slight change in the direction given to students (i.e., to answer the questions using their own perspectives and experiences).

*3.4. Data Analysis Procedures*

To understand teachers' perspectives of the curriculum, researchers initially analyzed the focus-group data using concurrent rounds of open coding for each school for a within-case analysis [45]. After the data were transcribed into transcripts (including teacher names) and field notes and drawings were added, two research assistants cleaned the data and placed it in a spreadsheet. Data were chunked into smaller, more meaningful parts (i.e., sentences). Descriptive coding [46] was used by each researcher to capture the experiences of teachers and their perspectives using the game-enhanced curriculum. A descriptive title (i.e., code) was used to label each talk turn. Initial codes included but are not limited to (a) task too easy or too hard, (b) scripting, (c) discussing thinking, (d) deficit language for students, (e) understanding game tasks, (f) tool use, and (g) characters in game. Two research assistants refined the codes and their meanings using comparison and collaborative work [47], which also built trustworthiness. Inter-coder agreement initially ranged from 75% to 90% for the data from each focus group. The process was then repeated to gain additional detail and specificity, which yielded two levels of codes. Categories were then identified by grouping codes together (i.e., axial coding). Inter-coder reliability across codes was 100% after collaborative work resolved disagreements [47]. Next, a cross-case analysis was conducted to identify the shared experiences of teachers who implemented the game-based curriculum [45]. Researchers then used the memos and collaborative work from the within-case analysis to guide this analysis. Similar and contrasting evidence across cases was identified, compared, and contrasted to create holistic descriptions of teacher perspectives of implementing the game-based curriculum. Combining focus-group data with memos ensures trustworthiness of the results [48].

To understand the teachers' adherence to the curriculum, researchers examined the fidelity checklists used to observe each teacher to calculate adherence and dosage information. For each area of the curriculum (i.e., previews, after game tasks), researchers counted how many items teachers enacted. They then divided that total by the total number of possible items, generating a percentage for each lesson component within each lesson as well as an overall adherence score for the lesson holistically. Researchers then repeated the process for all lessons observed for each teacher. To generate final percentages, researchers averaged all lesson adherence scores to obtain an average level of adherence for each teacher. Lesson timings were averaged to determine teachers' mean time spent teaching the intervention lessons (i.e., dosage). One-way ANOVAs were conducted to determine if teachers' mean adherence and dosage were significantly different.

Finally, to understand the extent to which students' fraction schemes and STEM interest change after participating in a game-enhanced intervention, researchers calculated normalized learning gains, which is used as an assessment of student knowledge of fractions and their STEM interest. Normalized learning gain (NLG) is the difference in post- and pre-test scores standardized by the maximum possible amount of increase (for learning gains) or decrease (for learning losses) from the pre-test score of the student. For fraction knowledge and STEM interest specifically, this metric helps create a fair comparison among students of different measured prior knowledge, as students who scored high on the pre-test were still capable of achieving high NLG scores since their maximum possible amount of increase is comparably lower than a student who scored low on the pre-test. NLGs and one sample *t*-tests were also used to evaluate responses to the S-STEM before and after the intervention. Univariate ANOVAs were also run to determine if differential program

effects (fraction and STEM interest NLGs) could be found across teachers, adherence levels, or dosage levels.

### 3.5. Merging and Final Interpretation

To gain a final interpretation of the data, trends in the data were identified, merged, and compared. First, researchers prepared a classical content analysis to quantify and identify trends in the focus group data by teacher. Researchers counted how many times each category and its indicators (i.e., strategies; consequences) were present in the data (i.e., coded talk turns) and then divided by the total number to obtain percentages to understand which category or subcategory were dominant for each teacher. Next, results of the classical content analysis were merged with each teacher's integrity data and their students' change in fraction schemes scores and change in S-STEM. Commonalities or divergences across these analyzed data were compared, and *t*-tests were used to determine any significant differences in the number of codes appearing in each category and subcategory for each teacher and yield a multifaceted understanding of teacher implementation, perspectives, and student success in the game-enhanced supplemental curriculum.

## 4. Results

### 4.1. Teachers' Integrity to the Curriculum

Our first research question addressed the extent to which teachers implement a game-enhanced supplemental fraction curriculum with integrity. To evaluate this question, we calculated teachers' average adherence and dosage for the supplemental curriculum, which are shown in Figure 1.

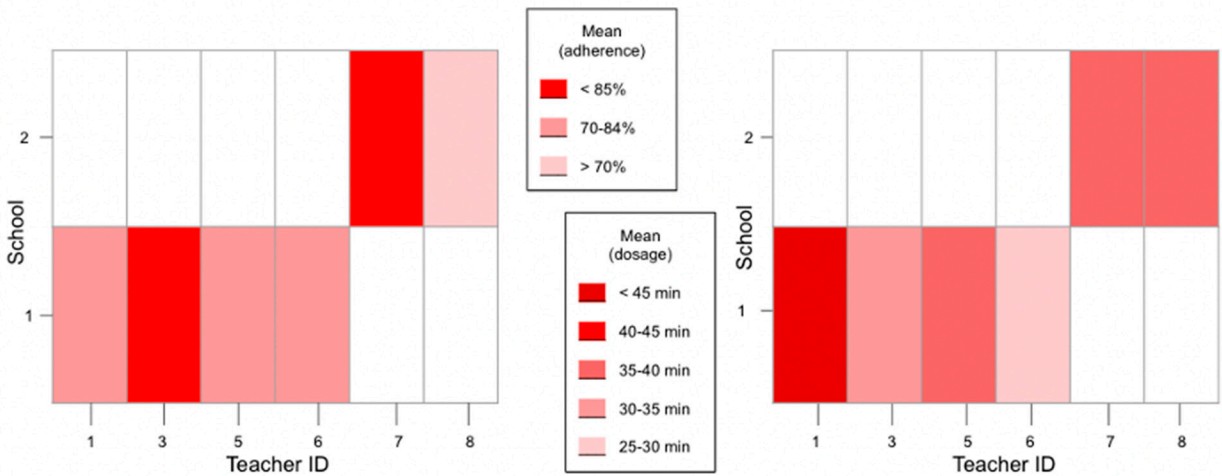

**Figure 1.** Teacher average adherence and dosage.

We can see that the highest adherence was observed in T3 and T7, with mid-level adherence observed in T1, T5, and T8. T6 had the lowest observed adherence. For dosage, three teachers (T5, T7, and T8) fell within 35–40 min of instruction, on average, per session. T3 had an observed average of 30–35 min per session. T1 was observed as having the highest dosage at an average of over 45 min. T6 was observed at the lowest average dosage per session of 25–30 min. A one-way ANOVA was conducted to evaluate if teachers' mean levels of adherence and/or dosage differed significantly from each other. The test showed a statistically significant difference in adherence ($F(5) = 9001.64$, $p < 0.001$) but not dosage. Teachers 3 and 7 had significantly higher adherence than teachers 1, 5, 6, and 8.

### 4.2. Teachers' Implementation Expereinces and Perspectives

Our second research question addressed the perspectives and experiences of teachers who implemented the supplemental game-enhanced curriculum. Results of our analysis

revealed three overarching categories: (a) Time, (b) Too Different, and (c) Too Difficult. We unpack each category and its subcategories below.

### 4.2.1. Time

For Time (see Figure 2), teachers spoke of and exhibited the phenomenon of the allotted time they expected to spend, with the program each day running longer than expected. Core scheduling and testing mandates emerged as an intervening context for this category.

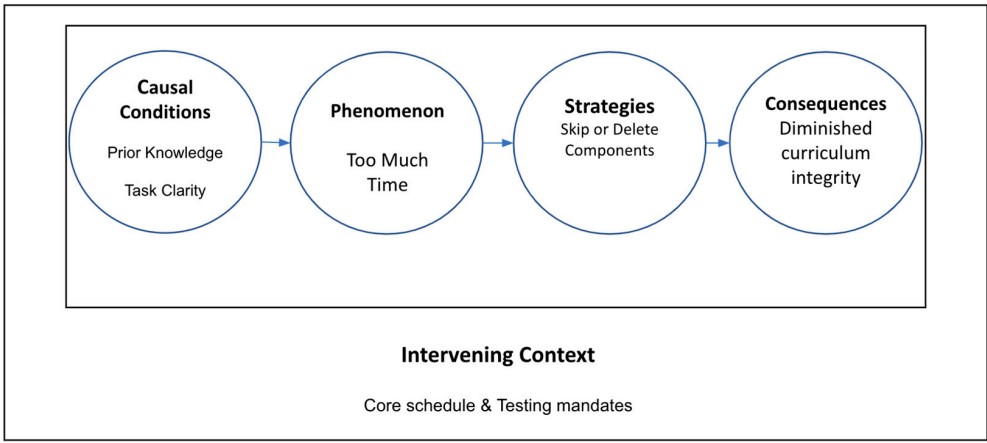

**Figure 2.** Time category.

The causal conditions named for the phenomenon included task structure or clarity (e.g., teacher and/or student understanding of what the task was asking) and prior knowledge (e.g., teachers felt that they had to do extra explaining of related ideas they felt students may not know in the curriculum task). Strategies teachers named to deal with the phenomenon of time include skipping or deleting parts of the curriculum or game time, with consequences being diminished curriculum integrity. Some teachers spoke of deleting game or curriculum time due to their deficit-based perceptions of their students' prior knowledge. A fourth-grade teacher commented:

> I know that like you know, when you're working with Tasha [game world five] on her and the map creating [game world five], like the rocket blast off, they were really like, it was a lot of thought for them. Their minds were stretched and they didn't like it. They are the corona kids [referring to students who took part in virtual instruction due to COVID-19 school closures] so for time we skipped or skimmed those games or lessons.

Other teachers spoke about prior knowledge in relation to their perceptions of the clarity of the after-game task, stating:

> I do think some of the questions were a little difficult for the kids to understand, like maybe they didn't understand what it was asking and then sometimes I felt like maybe I didn't understand what it was asking either. It took a lot of time! So I just skimmed it or I didn't say it.

A fourth-grade teacher spoke of altering the number string after game tasks, specifically, stating, "We're just so focused on getting the curriculum, you know our standards because everybody is so far behind that we weren't taking the time to do them properly [number strings]. So, I just ran through the problems". Finally, teachers spoke of deleting the curriculum because they perceived the task structure as repetitive, and that their students had already completed parts of the lessons at earlier points. A fifth-grade teacher said:

> Sometimes you know, like it [worked example] had justification on there and by the time we got to that point they had already pretty much justified it, so a lot

of the times that part was skipped because of that. We had discussed it, so you know I just didn't feel like they needed to take the time to discuss it again.

### 4.2.2. Too Different

Conversely, in the Too Different category (See Figure 3), the phenomenon motivating the situation was not that the game and curriculum concepts or teaching methods were taking too much time, but that they were very different from those that teachers and students were already using in their core curriculum. Teachers' instructional beliefs emerged as an intervening context in this category.

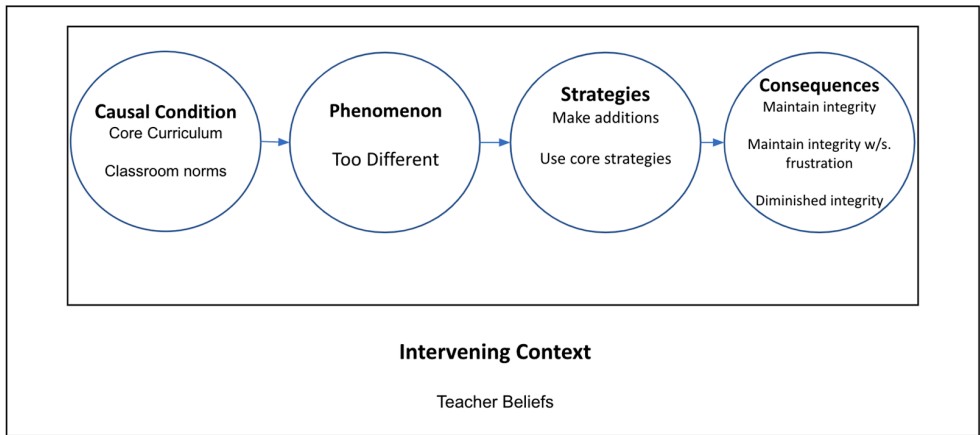

**Figure 3.** Too different category.

Classroom norms and beliefs about students were causal conditions for the curriculum being viewed as too different. For example, the supplemental curriculum promotes students to solve problems and have discussions with their peers about their strategies, while the core curriculum is based on direct or explicit instruction methods. Our analysis uncovered times at which these two causal conditions intersect. A fourth-grade teacher commented, "And I had a rougher group. They cannot work together. They, how do I want to put it, like they could, if they wanted to, but they didn't know how to talk to each other. Like I had to do a lot of prompting".

Strategies that teachers used to address the phenomenon differed across teachers. In some cases, teachers elected to continue to use the norms and teaching styles pushed by the core curriculum, which led away from the integrity of the game-enhanced curriculum. One fifth-grade teacher commented about the multiple models used in the game and after game tasks and explained:

> I think it's great to have them modeling it in different ways (e.g., number line), but our curriculum had us only using area models for fractions. I tried to do some extra research, [yet] I ended up just going back to modeling how to use the area model in most cases with my students, because that's what we've been doing.

Other teachers, however, made additions that are named in prior research as equity and asset-oriented [4], such as stating expectations (e.g., making statements about what they expected to see and hear as students shared ideas in pairs and in groups). A fifth-grade teacher described the emphasis on students' thinking as different from her core curriculum approach. However, instead of reverting back to teacher-led instruction, the teacher described adding in expectations and modeling a sample student discussion: "I would say things like, 'I should see your pencil moving', or 'I should be hearing discussion with your partner about your math thinking'. We also used the sentence stems you guys provided and mocked a conversation a few times". Approaches like these maintained the intent and integrity of the game-enhanced curriculum.

### 4.2.3. Too Difficult

Finally, in the category of Too Difficult (See Figure 4), the phenomenon was the perception of teachers that the tasks were too hard for their students. Confidence (perceived student confidence, as well as that of the teacher in their own abilities) emerged as an intervening context for this category.

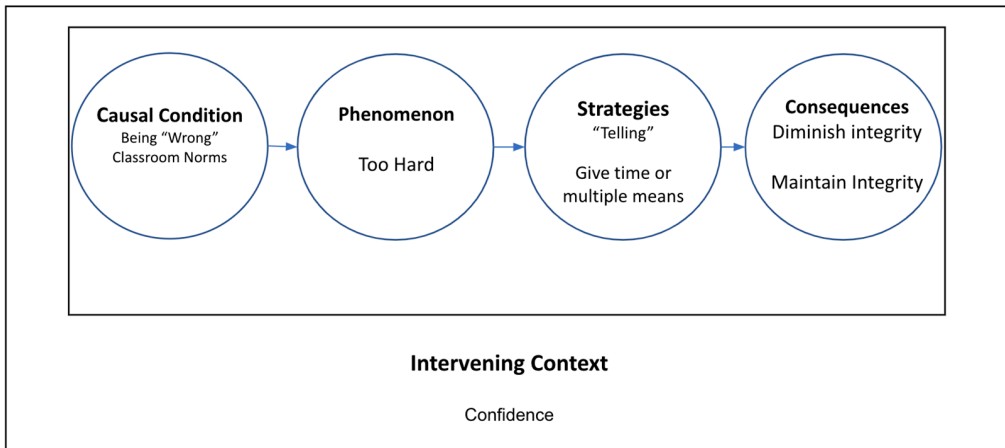

**Figure 4.** Too difficult category.

Causal conditions included ideas and norms around students "being wrong", with strategies again differing across teachers and leading to diminished or maintained curriculum integrity. For the latter (i.e., time and multiple means) teachers gave more time, additional means, or additional modalities for students to share their thinking about the tasks, such as allowing students to discuss their own reasoning (as opposed to the reasoning shown by the worked example) or allowing students to show a drawing or gesture, or write to show their reasoning (as opposed to speaking words). For example, in the number strings, teachers described including "turn and talks" before the whole group share outs and/or a whiteboard where students could draw out or model their thinking and honored those ways of expressing solutions as students shared. One fifth-grade teacher described altering a problem that a worked example was based on, asking students how they would solve it:

> As I would walk around you know you could hear them not arguing but discussing back and forth you know, 'no I don't agree with that' or 'yes, I think I think I've got this' back and forth. They really would have some very intense conversations about, you know, what's going on...'I think it's this way'; 'I think you can do it that way.' And I was amazed at some of the points that they came up with.

For the former (i.e., tell), teachers told students what to notice or think about the tasks. After describing a game level they perceived as too difficult for students, a fifth-grade teacher described circulating and telling students how to solve the game challenges. Further, they commented that the game should model or tell students how to solve the problem:

> I think it would be great if we could have the game solve it for them. Like, the game takes over and solves it for them and it tells them why it's true. And then, let them try the next one or give them a similar one and see if they're successful like that. I think that would work better for my students.

### 4.3. Student Learning

Our last research question addressed the extent to which students' fraction schemes and STEM interest changed after participating in the game-enhanced intervention. To address this question, we assessed student knowledge of fractions and their STEM interest

using normalized learning gains (NLG). Figure 5 shows students' NLG, which are average gains across all items on the fraction schemes tested across all students. The mean gain across all six teachers was 0.11, with a minimum score of $-1.00$ and a maximum score of 3.67.

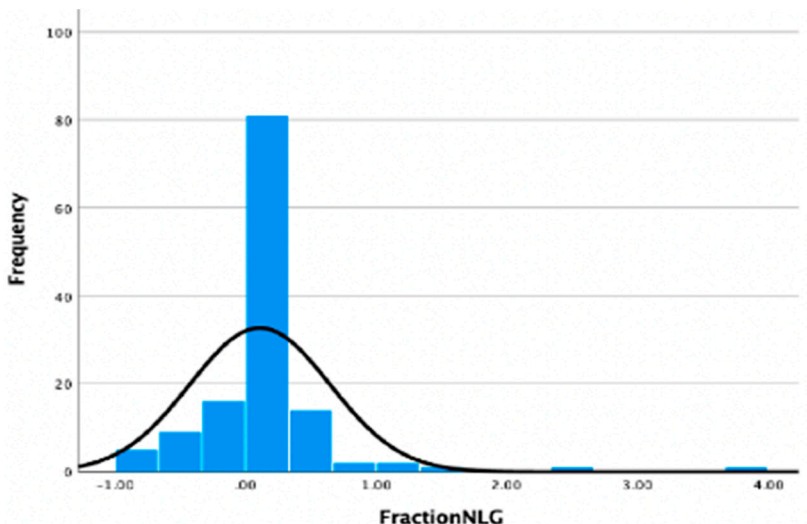

**Figure 5.** Overall fraction knowledge NLGs.

Figure 6 illustrates NLG broken down by teacher and school. As shown, teachers 2, 6, 7, and 8 had positive NLG (0.31, 0.09, 0.16, and 0.05, respectively), while T1 and T5 had negligible positive (0.01) and negative ($-0.01$) gains.

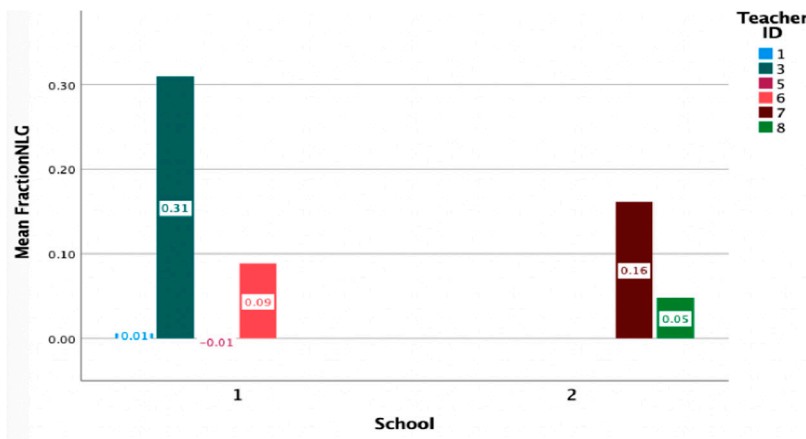

**Figure 6.** Fraction knowledge NLGs by teacher and school.

Figure 7 shows the NLG for students' self-reported STEM interest. These data illustrate average gains across all items on the S-STEM questionnaire across all students.

The mean gain across all six teachers was 0.17, with a minimum individual student score of $-0.57$ and a maximum student score of 3.40. Figure 8 illustrates NLG for STEM interest broken down by teacher and school.

As shown, teachers 3, 5, 6, and 7 had positive NLG (0.54, 0.12, and 0.31, respectively), while T1, T6, and T8 had small negative ($-0.06$, $-0.03$, and $-0.02$, respectively) gains.

Univariate ANOVAs, using fraction pre-test as a covariate due to unequal group sizes, were run to determine if the NLG across teachers, different levels of adherence, or levels of dosage were significantly different for fraction knowledge or STEM interest. Results revealed no significant difference in NLG based on any fixed factor. However, teachers who demonstrated higher adherence to the game-enhanced curriculum and had at least

the recommended levels of dosage or five minutes over generally displayed higher fraction and STEM interest NLG (see Figures 9–12).

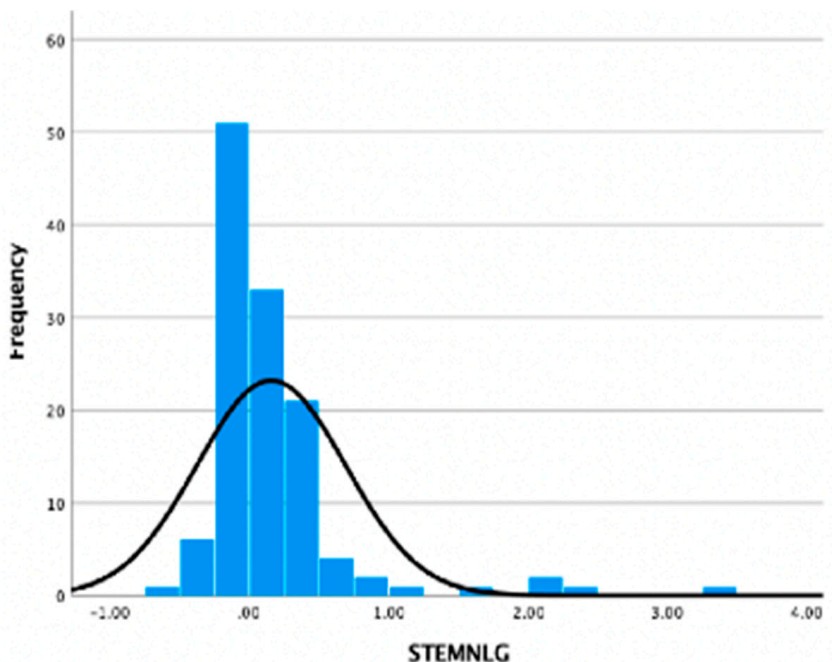

**Figure 7.** Overall STEM interest NLG.

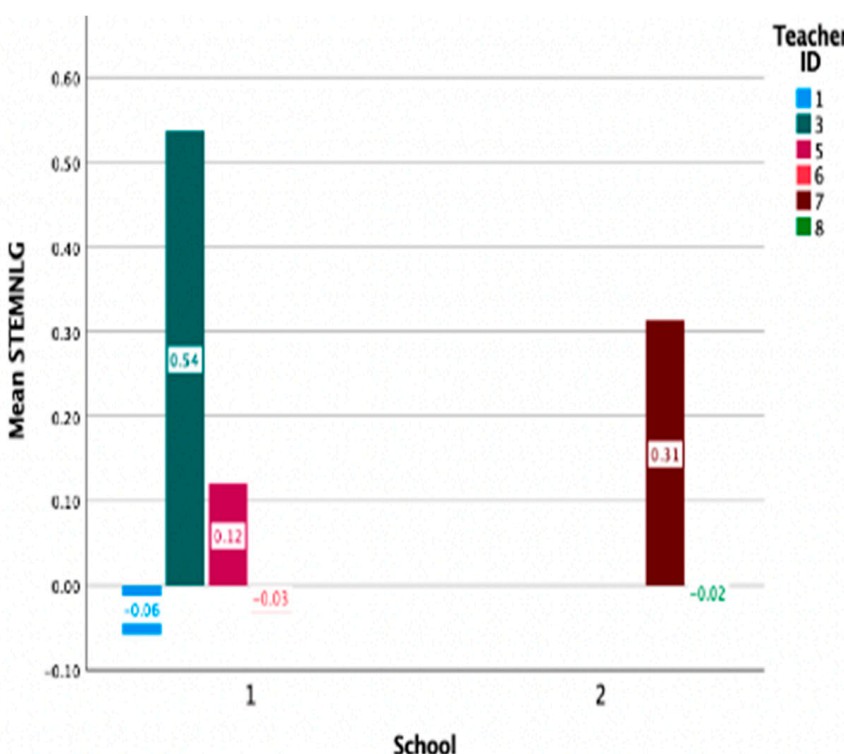

**Figure 8.** Overall STEM interest by teacher and school.

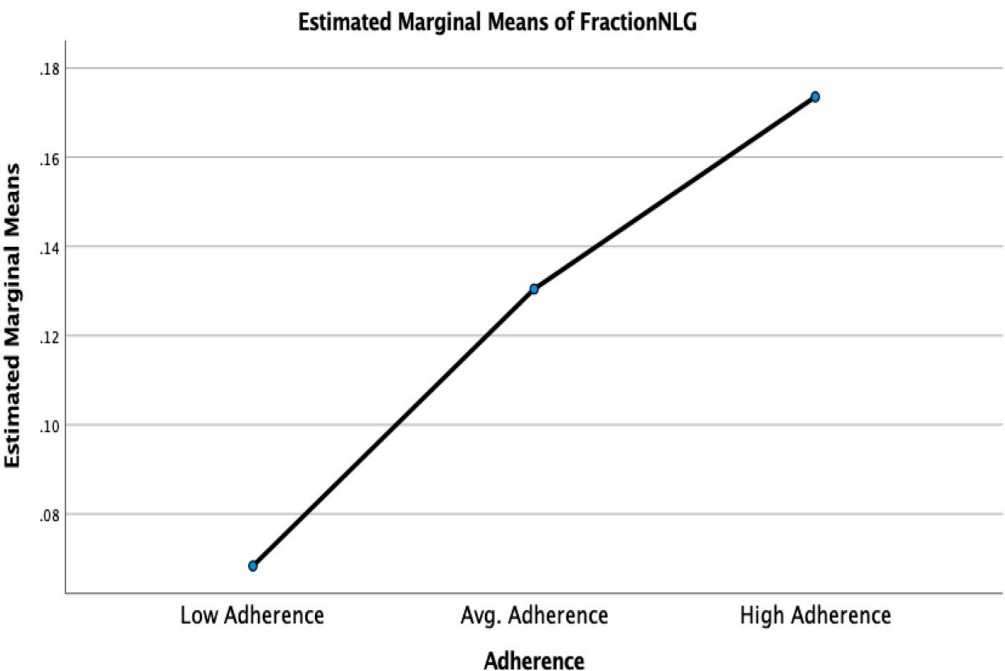

**Figure 9.** Adherence and Fraction NLGs.

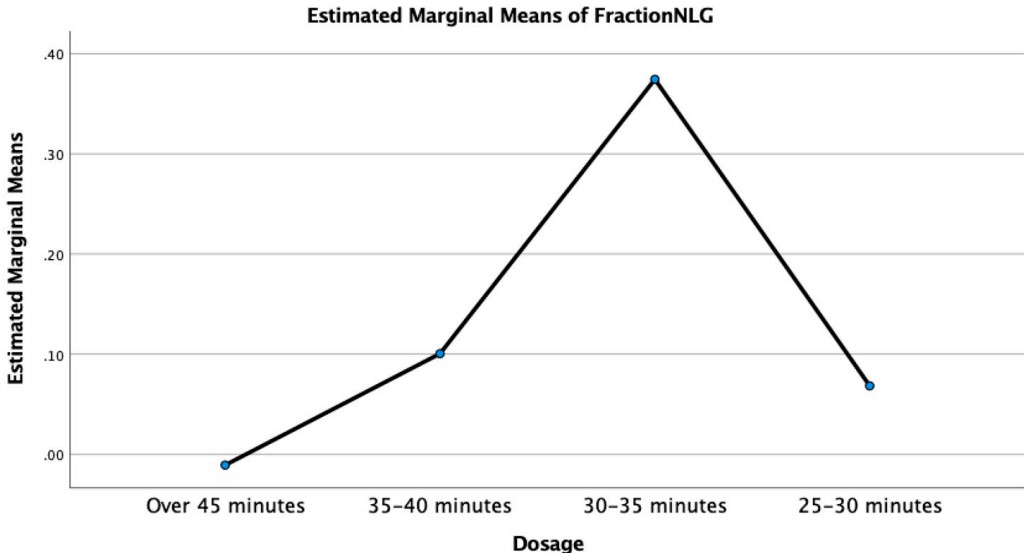

**Figure 10.** Dosage and Fraction NLGs.

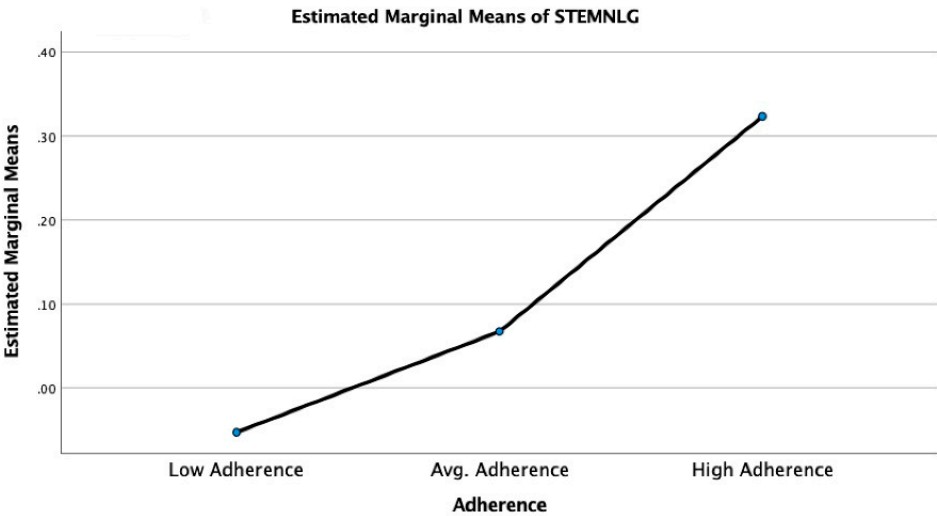

**Figure 11.** Adherence and STEM NLGs.

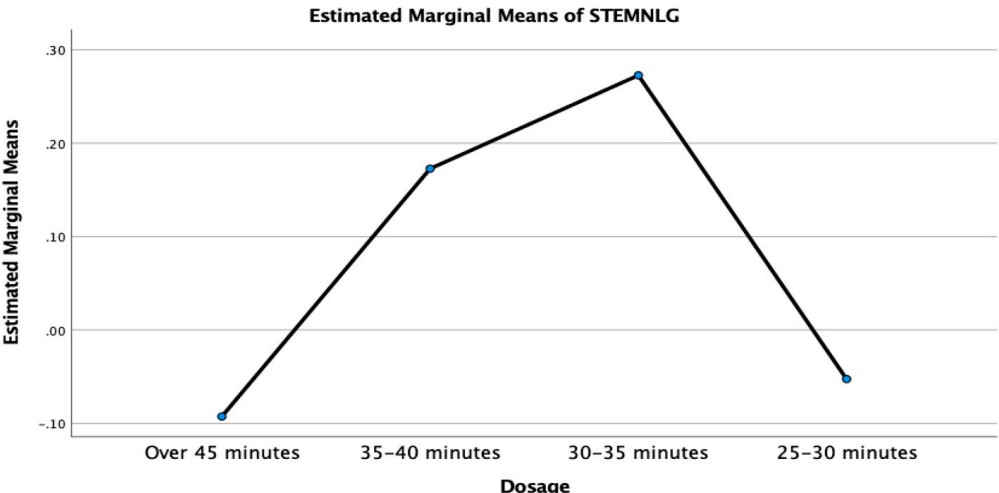

**Figure 12.** Dosage and STEM NLGs.

### 4.4. Merging and Interpretation

Predominant qualitative strategy codes that fall under each phenomenon are listed by teacher in Table 2.

In response to the phenomena of time, too different, and too difficult, teachers 1 and 5 used predominantly "delete", "use core", and "tell" strategies. Conversely, teachers 3 and 7 used predominantly "skip", "make additions", and "give more time or multiple means" strategies in response to the experienced phenomena. Teacher 6 displayed predominantly "delete", "make additions", and "tell" strategies, while teacher 8 predominantly utilized "delete", "use core", and "give more time or multiple means" strategies. Comparing the quantified focus group trends to teachers' delivery of the program gives context to the integrity and dosage scores and yields important takeaways regarding the feasibility of implementing the game enhanced program and the effects on student outcomes.

**Table 2.** Focus Group Category/Subcategory by Teacher, Fidelity, and Student Fraction/STEM Changes.

| Teacher | Pre-Post Tests | | Integrity | | Focus Group Predominant Code Presence | | |
| | Learning Gains | | | | | | |
| | Fraction | STEM | Adh. | Dos. | Time | Too Different | Too Difficult |
|---|---|---|---|---|---|---|---|
| 1 | 0.01 | −0.06 | Mid | <45 min | Delete | Use Core | Tell |
| 3 | 0.31 | 0.54 | **High** | **30–35 min** | Skip | Make Additions | More Time/Mult. Means |
| 5 | −0.01 | 0.12 | Mid | **30–35 min** | Delete | Use Core | Tell |
| 6 | 0.09 | −0.03 | Mid | 25–30 min | Skip | Make Additions | Tell |
| 7 | 0.16 | 0.31 | **High** | **30–35 min** | Skip | Make Additions | More Time/Mult. Means |
| 8 | 0.05 | −0.02 | Low | 35–40 min | Delete | Use Core | More Time/Mult. Means |

* BOLDPRINT = highest average learning gains in fractions or STEM interest.

Specifically, qualitative nuances in teachers' perceptions of time, alignment of the program with typical instruction, and views about students' capacity to participate in the supplemental program impacted program delivery in important ways. Teachers who made additions or provided more time and multiple means of accessing the program had higher adherence and greater increases in student learning and interest in STEM compared to teachers who used other strategies to address perceived issues of time, difficulty, or alignment of the game-enhanced program with core instruction. Teachers who chose to remove opportunities for student thinking (i.e., "tell"), revert to core instruction (i.e., "use core"), or both in response to issues of time, program difficulty, or alignment with core instruction, saw lower or even no changes in students' fraction thinking and STEM interest. Therefore, we conclude that teachers' integrity to the curriculum approach, alongside asset-oriented additions, contributed to improvement in students' fraction knowledge and STEM interest.

## 5. Discussion

The purpose of this study was to determine how teachers perceived and implemented a game-enhanced supplemental curriculum for fractions, a foundational STEM and ICT content area. We investigated whether teachers implemented a game-enhanced supplemental fraction curriculum with integrity, their experiences and perspectives after implementing the game-enhanced fraction intervention in their classrooms, and the extent to which students' fraction schemes and STEM interest changed after participating in a game-enhanced intervention. Prior research suggests that game-enhanced mathematics interventions are a way for teachers to promote access to prior knowledge, improve engagement, and empower students with disabilities by bolstering their learning outcomes [4–10]. Yet, the extent to which teachers used games interacts with several factors, such as teachers' beliefs about knowing and learning [11], the fit of games with existing curricula [31,32], and time [35]. Results of the current study add to this literature in several ways.

First, results of this study connect teachers' actualized implementation of a game-enhanced fraction curriculum to their perceptions, beliefs, and, ultimately, their students' learning outcomes. Previous studies illustrate that teachers will not implement digital games that they do not see as directly instructing students or those in which they themselves had not had the chance to learn about or fully understand. Yet, in our study, teachers' descriptions of the phenomenon of time were more about task structure or clarity, and also teachers' perceptions of students' prior knowledge. Teachers who reported deleting or skipping parts of the curriculum described time in ways that highlighted their perceptions of students' abilities or understanding ("Their minds were stretched and they didn't like it".; "We had discussed it, so you know I just didn't feel like they needed to take the time to discuss it again".). Combined with other strategies, such as telling or using pedagogical strategies from core instruction, these teachers implemented the curriculum with lower levels of integrity and saw little to no learning gains or changes in STEM interest for their students. Thus, the interaction of teachers' [deficit] beliefs about students and their propensity to take up newer instructional approaches warrants more research.

A second and related contribution to the literature is the documented strategies that teachers used when faced with other phenomena previously described in research on the use of digital math games, such as curriculum fit [31,32], the need to use direct instruction [28,32], or increased math performance [29]. On the one hand, some teachers who were faced with perceived differences between the supplemental curriculum and their core curriculum, or perceived difficulty of the supplemental curriculum, used strategies that added to (as opposed to took away from) the program in asset-oriented ways. For example, teachers added time, opportunities for students to rehearse their thinking, or made curriculum expectations explicit. Overall, these teachers also implemented the curriculum with higher levels of integrity and had higher student outcomes. Thus, examining interactions with teachers' asset-oriented views of students and how to promote them to support integrity in implementing curriculum innovations warrants further examination.

On the other hand, the results of this study also show that other teachers, in the face of the same implementation issues, chose to delete the core ideas of the curriculum, or change it to better align with their core instruction. As earlier noted, several of the teachers' coded statements and strategies seem to reveal not only misalignments between the supplemental game-based approach and their current practice but also their beliefs about instruction, mathematics, or their students. Prior research that examined students' outcomes and perceptions [6] suggested that students appreciated the curriculum for the agency they experienced within it and even wanted additional opportunities to see and hear themselves and their ideas in the curriculum. Yet, if teachers' own beliefs about instruction do not match, they may not provide such opportunities, even given a curriculum that provides it [31,32].

A third contribution relates to the open question regarding links between a multitude of teacher beliefs (i.e., digital games, student potential, math instruction), their knowledge of student thinking, the realities of classroom pressures, and, ultimately, the potential for innovative curricula to support instructional change that has the potential to bolster student achievement. It may be that tangible factors, such as time or alignment with schools' predominant curriculum approaches, impact teachers' use of supplemental game-based programs more, less, or the same amount as their beliefs. Prior research illuminated the role of factors such as time, cost, and curriculum focus to increase standardized test scores which led to hesitation in implementing contemporary game-based approaches, even when beliefs about the use of and/or potential of games were positive [33]. Yeo et al.'s research [32] showed direct relationships between attitudes surrounding games and their use in the classroom; teachers' perceptions of fit and time mediated those relationships.

The findings from this study support, and also extend, prior work in that teachers were all subject to the same implementation concerns yet employed different strategies to address them in ways that impacted not only their use of the game-based curriculum but also student outcomes in the classroom. Arguably, the differences in strategies are the result not of the teachers' beliefs or attitudes about games, but their beliefs about mathematics instruction and pedagogy [49], or students' potential in curricula that front mathematical thinking as opposed to teacher demonstration or teachers' expertise in student-centered curriculums [11]. More research is needed to investigate these relationships in depth.

## 6. Limitations and Future Research

Limitations of this research need to be acknowledged. The first limitation is due to the nature of the research design; the small sample of teachers and students make the findings of this work limited to this group of teachers. We do not make claims that the results extend beyond the experiences and context of the current study. A second limitation is the self-reported nature of our data. We used focus groups with teachers to understand their perceptions of the game-based curriculum. While we did observe teachers to understand their curriculum integrity, we did so in a way that did not allow us to capture actualized, real-time changes teachers did make or why they made them. Future research might include interviews or more nuanced observation tools that allow for this information to be

collected and analyzed so results can be compared with the findings of the current study. A related third limitation is that, while we make conjectures regarding relationships between teachers' beliefs, their implementation of the curriculum, and students' outcomes, we did not use a measure of teacher beliefs in this study to document the themes that emerged from our analysis of the focus groups. While our grounded theory and analysis support our claims about teachers' beliefs, more data and research are needed to formally test our assertions about the relationships between, say, teachers' beliefs about students and their use of innovative curricula in the mathematics classroom.

## 7. Conclusions

In this work, we learned that a game-enhanced supplemental curriculum has the potential to positively impact students' fraction knowledge and STEM interest. Teachers in this study who implemented the program with integrity often expressed asset-oriented beliefs about students that seemed to support teachers to stay consistent with the intended instructional approach. Program revisions are planned to produce increased curriculum integrity, specifically in relation to how teachers see students' potential to learn from their own reasoning, their knowledge of ways to adapt game-enhanced curriculums to keep student thinking at the forefront, and the ways in which they receive support to do so in real time. Given the continued need to produce STEM interested and ICT prepared students, the need for continued work and research in these areas is paramount.

**Author Contributions:** Conceptualization, J.H.; methodology, J.H., M.T., B.B. and A.D.; software, J.H. and M.M.; formal analysis, J.H., M.T., B.B. and A.D.; investigation, J.H., M.M. and M.T.; resources, J.H., M.M. and M.T.; data curation, J.H.; writing—original draft preparation, J.H., K.W.-A.; writing—review and editing, J.H., K.W.-A., K.H., M.M., M.T.; visualization, A.K.; supervision, J.H., M.M. and M.T.; project administration, J.H., M.M. and M.T.; funding acquisition, J.H., M.M. and M.T. All authors have read and agreed to the published version of the manuscript.

**Funding:** This research was funded by the US National Science Foundation grant number 1949122.

**Institutional Review Board Statement:** The study protocol was approved by the Institutional Review Board (or Ethics Committee) of University of Central Florida (14238A03, 11 December 2021).

**Informed Consent Statement:** Informed consent was obtained from all subjects involved in the study.

**Data Availability Statement:** Not applicable.

**Conflicts of Interest:** The authors declare no conflict of interest.

## Appendix A. Sample Checklist for Worked Example

**Worked Example (20 minutes)**

☐ (1 minute) Show Slide 4. Say, *"The wind is flowing to three turbines. Here is the result of Bunny's estimate for one share of the wind repeated three times. Bunny decides to make the next estimate shorter."*

☐ (4 minutes) Show Slide 5. Say, *"I want you to explain to yourself why making the next estimate shorter doesn't work. Here are some questions to help you. Use at least one 'what' question and one 'why' question to write your response."*

☐ (5 minutes) Show Slide 6. Say, *"Now it is time to discuss your explanation with a friend. One person can explain while the other responds. Use the thought bubbles to help you explain and respond. Then, switch roles."*

    ☐ Encourage students to use the thought bubbles to help them talk with each other.

    ☐ Acknowledge students who look like they have something to say (e.g., "___, you have an idea. What is it?")

☐ (3 minutes) Show Slide 7. Say, *"Now is the time for you to justify your arguments. To do that, add more evidence. Evidence can be pictures, numbers, or more words that support your points. You can also respond to questions your friend had about your explanation. This is new, so try for a few minutes and then we'll check in about your ideas for justification."*

☐ (5 minutes) Show Slide 8. *"Let's talk about some ways we could justify our arguments using pictures, numbers, and/or words:"*

    ☐ *"Let's say you are trying to support the claim that Bunny should have made his next estimate longer. How could we give evidence? How about drawing a picture? What might you draw to show that?"* (take ideas, validate ideas, click to show a possible drawing).

    ☐ *"How might you use numbers to support the claim that it should have been longer?"* (take ideas, validate ideas, click to show a possibility).

    ☐ *"We needed the picture and some words to show more evidence with numbers. What else might we say with words to explain that bunny needs the next estimate to be longer than the original?"* (take ideas, validate ideas, click to show a possibility).

    ☐ *"Now you have seen a few different ways to justify your arguments. Continue with the justification you started."* Allow students 3 more minutes of work time.

☐ (2 minutes) Show Slide 10. Ask students to make the next estimate on their own. Collect responses.

If any are unchecked, explain:

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
