# Peer review of "Elementary Teachers’ Perceptions and Enactment of Supplemental, Game-Enhanced Fraction Intervention"

_education, doi:10.3390/educsci13111071_

Round 1

Reviewer 1 Report

The article titled "Elementary Teachers’ Perceptions and Enactment of Supplemental, Game-Enhanced Fraction Intervention" study examines the perceptions and implementation of a supplementary intervention for fractions, enhanced with educational games, as perceived and enacted by elementary school teachers. The study presents novel and significant findings pertaining to the utilization of games in the instruction of mathematics. Digital Game-Based Learning (DGBL) is frequently hailed as an innovative approach to education, and the present study presents empirical evidence regarding the outcomes of implementing a fraction game. The research also contributes to the expansion of discoveries pertaining to the STEM field.

The literature review exhibits a high level of detail and sufficiency, while the framework employed for analyzing the findings is also commendable. The methods are thoroughly elucidated, along with a comprehensive description of the data analysis process. The findings exhibit a high degree of clarity and coherence.

Upon careful examination of the text, no instances of logical inconsistencies or disagreements were identified. Therefore, it is recommended that the work be published in its current form.

Author Response

Thank you for your positive comments and thoughtful review!

Reviewer 2 Report

An interesting topic and idea.

It should be clarified why the authors consider this sample (by the number of teachers and students) representative, which sources can be referred to regarding the representativeness of the sample.

Is it possible to answer the questions of the study (for the purpose of the study) using a sample of this volume?

You may need to work on pictures (single presentation style)

ok

Author Response

  1. An interesting topic and idea.

Thank you.

2.  It should be clarified why the authors consider this sample (by the number of teachers and students) representative, which sources can be referred to regarding the representativeness of the sample.

The sample is purposive and thus representative of this group of teachers in this particular setting/region/state, etc.  We do not claim that the sample is representing anything beyond that.  We list this information in the limitations section.

3.  Is it possible to answer the questions of the study (for the purpose of the study) using a sample of this volume?

To our knowledge, mixed methods studies do not require a sample size, as we are not trying to generalize results of this study.  Instead, the results are relevant to this group of teachers.  We list this in the limitations section.

4.  You may need to work on pictures (single presentation style).

I'm not clear what the actionable recommendation is with this comment.  Can you say more?

Reviewer 3 Report

This article was nicely written and structured well. It's clear that the authors worked very hard on it (and the study). I hope you find these comments to be helpful as you continue to do research on games and elementary education. 

I had three main concerns about the article: a. implementation measurement; b. contributions to the field; and c. qualitative methodology.  

Implementation measurement: there are a range of approaches to implantation measurement in the field as well as a range of frameworks that organize the possible influences on implementation. None were referenced. The authors didn’t include the Appendices so I had no idea what the implementation “checklist” looked like which left me unable to determine if the checklist was a good measure of implementation.

However, I do question the implementation analysis of creating percentages and then averages. Typically, the “parts” of instructional materials are not of equal importance (and that method makes them of equal importance) so here, the reader is left only with a general “how much” and nothing more (e.g. which parts were used, which parts were skipped, were the skipped parts core to the curriculum or were they less important?) Further, there is an assumption that complete “adherence” is best. However, one should expect a teacher to make adaptations based on their students and contexts. At one point the authors referenced a teacher who skipped something because her students already knew it. Skipping the materials seems like a reasonable modification but it wasn’t clear what the authors point of view on this was. People who measure implementation have other terms than "destroy."  There are  “principled” adaptations (that make it better) and “fatal” adaptations (that make it worse).

All in all, the implementation measurement appeared to be less rigorous than it could have been (though I can’t fully form an opinion without seeing the checklist).  The reader is left only with knowing that some teachers did more and some did less but nothing more. 

 So, I suggest that for any future studies, the authors look into the literature on implementation measurement.

 Also, I do want to state the obvious issue (as did the authors) – that the study was only 6 teachers in two schools. There was no accounting for the teachers’ preexisting practices and approach. It’s very possible that the students in the classrooms with more adherence did well simply because they had a better teacher.

Findings are known:  Regarding time - the lack of time is a long-standing and well known problem that has been established in practice and research. So, this finding was quite predictable .  In fact, I was as surprised to see that the expectation was 35 minutes 3 times a week – that is not trivial in the week of an elementary school teacher. Similarly, it is common for elementary teachers to talk about the pressures of meeting standards and testing and the impact it has on their classroom practice.

Too Different: In the literature on implementation and influences on implementation, this would be called “fit” with the curriculum. This is one among many other possible influences such as perceived importance, ease of use (the “too hard” category), value for the students, adaptability, complexity and many more. Other influences relate to the teacher themself such as self-efficacy, interest, persistence and as mentioned in the article, beliefs about students. Even though this research focused on a game-based innovation, the influences on implementation are still the same and therefore, are known. It would have been more interesting if the authors had researched influences that were unique for game-based innovations.

Qualitative Methodology:  My first instinct was to ask why the researchers opted for focus groups rather than interviews since there were only 6 teachers and in one school the “group” was only 2 teachers. Had there been interviews, there at least would have been 6 transcripts to analyze instead of 2 (if indeed transcripts were used – there is no mention of transcripts so it could have been the notes the researcher took?).

The qualitative methodology description was rigorous, but actually too rigorous to analyze 2 45-minute focus group transcripts (if there were transcripts). I also want to know if the transcripts (or notes) were labled with the teachers’ names? Focus groups can be very difficult to transcribe (and even document).  See additional comments/questions below: 

a) It was not clear if the researchers actually made observations and if the checklist data was from the observations or audio recordings. The only mention of what could have been observation data was possibly the mention of  “field notes” added to the data. What were these? There was no mention of an observation protocol (unless it was simply the checklist) but then, what could the "field data" have been?  More concretely, were the checklists completed in both observations and with the audio tapes? On line 185, there is a description of a random number generator choosing a lesson and then the lesson was “observed…provided the audiotape was of acceptable quality.” So, were these lessons observed or not? If yes, what is the reference to the audio recording? Also, there is a mention of “memos” on line 320 but no mention of what the memos were.

b) What does in mean that the qualitative data was “cleaned” and put into a spreadsheet? What were the columns and rows?

c) What is “descriptive coding within a grounded theory?” As I'm sure the authors know, grounded theory is an inductive approach focused on the creation of a theory from series of coding processes. There was no theory developed here and when I looked at the reference, it was about the S-STEM instrument, not qualitative research.

d) What were the “more meaningful parts” that were “chunked?” (I don’t know what a “talk turn” is and couldn’t find that term anywhere in a Google search. A talk turn seems like something one might look for in discourse analysis but in this context, I couldn't speculate. 

Below are some smaller and editorial notes:

·      Line 29 – Omit “overly.”

·      Line 151 – This is a long, difficult to understand sentence.

·      Why weren’t teachers compensated for their time?

·      Line 169: I didn’t know what it meant to deliver the curriculum through role playing, rotating between roles and student roles. 

·      Line 179:  – The citation [37] does not have to do with tech-based interventions so this must have been mis-cited.

·      Line 222: The lesson is really 40 minutes if taught at the highest end of the recommended time.

·      Line 245: This paragraph was hard to follow. It would be helpful to break up the text by game replay, worked example, and after game play.

·      Line 270: Was this a focus group or a semi structured group interview? It’s unusual to describe a focus group as asking “each question one at a time.”

·      Line 305: What were the drawings?

·      Line 33: How was the holistic score determined?

·      Line 381: Typo in heading

Additional comments: 

Passive voice and some long sentences made it a little difficult to process in scattered places.

I also took note that asking teachers to commit to doing lessons 3 times a week for 35 minutes is not trivial at the elementary level. So, although 6 teachers agreed to do so, this innovation is much less feasible on a larger scale.  

Author Response

1. Implementation measurement: there are a range of approaches to implementation measurement in the field as well as a range of frameworks that organize the possible influences on implementation. None were referenced. The authors didn’t include the Appendices so I had no idea what the implementation “checklist” looked like which left me unable to determine if the checklist was a good measure of implementation.

The study was looking at small scale feasibility through the lens of fidelity of implementation and students outcomes. Bos et al's (2022) approach is cited and taken up in this work as a framework to examine fidelity of the approach.  This term is commonly used in special education research and development.  

We've now included the appendix in the paper.  Please note the included appendix is a sample and lesson specific.

2. However, I do question the implementation analysis of creating percentages and then averages. Typically, the “parts” of instructional materials are not of equal importance (and that method makes them of equal importance) so here, the reader is left only with a general “how much” and nothing more (e.g. which parts were used, which parts were skipped, were the skipped parts core to the curriculum or were they less important?)

We understand your perspective, yet to us each component was equally as important in this phase of feasibility.  So, each part was accounted for equally.  

3. Further, there is an assumption that complete “adherence” is best. 

In this context, yes, we were looking to see if teachers used the approach as intended and its relation to student outcomes.  It is common in special education to use adherence as one measure of fidelity.  

4.  However, one should expect a teacher to make adaptations based on their students and contexts. At one point the authors referenced a teacher who skipped something because her students already knew it. Skipping the materials seems like a reasonable modification but it wasn’t clear what the authors point of view on this was. People who measure implementation have other terms than "destroy."  There are  “principled” adaptations (that make it better) and “fatal” adaptations (that make it worse).

We appreciate that teachers may certainly change the curriculum to suit their purposes, and we agree with you.  However, in this particular stage of our research, which was assessing feasibility as conceptualized by fidelity of implementation, our intent was to simply report whether something was skipped, changed, or done as intended.  We do not make a value judgement (better, worse)...we only report whether something was implemented or not.  

5.  All in all, the implementation measurement appeared to be less rigorous than it could have been (though I can’t fully form an opinion without seeing the checklist).  The reader is left only with knowing that some teachers did more and some did less but nothing more. 

We appreciate your perspectives, yet from our stance, the reported information matches the research questions under study (e.g., if the approach was used as intended, teachers' perspectives regarding the curriculum, and student outcomes).  A sample checklist is attached in the revised paper.

6. I suggest that for any future studies, the authors look into the literature on implementation measurement.

We will certainly do that as we develop other research questions that address implementation in ways different than the focus of the current study.  However, this particular stage of our research has the goal of assessing feasibility as conceptualized by fidelity of implementation.  

7.  Also, I do want to state the obvious issue (as did the authors) – that the study was only 6 teachers in two schools. There was no accounting for the teachers’ preexisting practices and approach. It’s very possible that the students in the classrooms with more adherence did well simply because they had a better teacher.

We appreciate that suggestion and will build other information about practice or approach into future iterations of our work.  In this study, we did not collect that information so we do not have it to share in this paper.

8.  Findings are known:  Regarding time - the lack of time is a long-standing and well known problem that has been established in practice and research. So, this finding was quite predictable.  In fact, I was as surprised to see that the expectation was 35 minutes 3 times a week – that is not trivial in the week of an elementary school teacher. Similarly, it is common for elementary teachers to talk about the pressures of meeting standards and testing and the impact it has on their classroom practice.

Best practices for math interventions are at minimum at this level.  We acknowledge that time, standards, and testing are documented in the literature (and that our study adds to and supports prior research here).  We also point out that novel contributions exist in how factors such as "time" play out to support (or not) teachers' use of novel technologies. For example, in our study, teachers’ descriptions of the phenomenon of time were more about task structure or clarity and also teachers’ perceptions of students’ prior knowledge. The interaction of teachers’ [deficit] beliefs about students and their propensity to take up newer instructional approaches is novel and warrants more research, especially in relation intervention research and the use of new practices to bolster math learning for students who are experiencing math difficulty.  

9.  Too Different: In the literature on implementation and influences on implementation, this would be called “fit” with the curriculum. This is one among many other possible influences such as perceived importance, ease of use (the “too hard” category), value for the students, adaptability, complexity and many more. Other influences relate to the teacher themself such as self-efficacy, interest, persistence and as mentioned in the article, beliefs about students. Even though this research focused on a game-based innovation, the influences on implementation are still the same and therefore, are known. 

We appreciate your comments.  We argue that the fact that findings from new programs reflect findings documented in other literature does not disqualify its importance - it extends prior literature in several ways.  First, our work documents that fact that these same issues show up in technology enhanced programming and that there are connections to outcomes for students.  Researchers and practitioners may assume technologies or technology enhanced programs  as a way to increase outcomes for, say, students who are experiencing math difficulty without considering perspectives of the teachers who use them. This study also extends the literature by showing possible impacts of teachers' perceptions of issues like time or or fit and the connection to student outcomes.  

10.  Qualitative Methodology:  My first instinct was to ask why the researchers opted for focus groups rather than interviews since there were only 6 teachers and in one school the “group” was only 2 teachers. Had there been interviews, there at least would have been 6 transcripts to analyze instead of 2 (if indeed transcripts were used – there is no mention of transcripts so it could have been the notes the researcher took?).

We used focus groups as opposed to interviews at the request of the partner schools. The paper describes the data sources on pages 6 and 7, which included transcribed sessions, field notes, and drawing schemas of the interactions.

11. The qualitative methodology description was rigorous, but actually too rigorous to analyze 2 45-minute focus group transcripts (if there were transcripts). I also want to know if the transcripts (or notes) were labeled with the teachers’ names? Focus groups can be very difficult to transcribe (and even document).  See additional comments/questions below.

We are not aware of any best practice in qualitative research that states not use within and between case analyses to understand a research question involving focus groups.  Teacher names were included in the transcriptions.  

12a) It was not clear if the researchers actually made observations and if the checklist data was from the observations or audio recordings.

We describe this on page 5 of the paper.

12b) The only mention of what could have been observation data was possibly the mention of  “field notes” added to the data. What were these?

Observation data collection was described on page 6 and included below.  There is no mention of field notes.

"Quantitative data collection. To gauge teachers’ integrity to the curriculum, a research assistant listened to each selected lesson tape, while completing a checklist (see Appendix) to identify the essential points the teacher implemented. The checklists are a point-by-point accounting of each statement and teacher action required by the curriculum. The research assistant checked off each item as it was enacted by the teacher as captured by the recording. The research assistant also recorded at the top of each checklist the timing of each lesson (i.e., the amount of time the teacher spent in each lesson segment). They also added descriptive information into sections embedded into the checklist areas of the curriculum that teachers added to, deleted, or changed."

12c) There was no mention of an observation protocol (unless it was simply the checklist) but then, what could the "field data" have been?  More concretely, were the checklists completed in both observations and with the audio tapes?

We answered these questions in responses above.  We clearly described the data collection methods (e.g., checklist for observation) and procedures for using it with data - the answers to these questions are already in the paper.

12d) On line 185, there is a description of a random number generator choosing a lesson and then the lesson was “observed…provided the audiotape was of acceptable quality.” So, were these lessons observed or not? If yes, what is the reference to the audio recording? Also, there is a mention of “memos” on line 320 but no mention of what the memos were.

I'm not sure what the question is in this comment.  Can you reword it?  I think the sentence states that the lesson was observed if they audio was quality.  That means that it wasn't if the audio was poor, and we discussed what the procedure was for selecting an alternate tape.  I cut and pasted that paragraph below:

"Once a lesson was identified for review by the random generator, that lesson was observed for all teachers using a checklist (see the Appendix and data collection procedures), provided the tape was of acceptable quality to listen to the lesson in its entirety. If it was not, then a lesson immediately preceding or following the selected lesson was observed."

12e) What does in mean that the qualitative data was “cleaned” and put into a spreadsheet? What were the columns and rows?

Once the data is fully transcribed it may be necessary to ‘clean’ the data. This is especially true of interview transcripts which tend to be littered with asides and repeated points. Removing these from a transcript will make the data easier to work with. It is important to note that cleaning does not involve any change to the wording used, only the removal of any parts that are not to do with the nature of the focus group or the removal of words that are commonly used in informal speech but are not used for more formal written reports. 

12f) What is “descriptive coding within a grounded theory?” As I'm sure the authors know, grounded theory is an inductive approach focused on the creation of a theory from series of coding processes. There was no theory developed here and when I looked at the reference, it was about the S-STEM instrument, not qualitative research.

The reference was not properly traced to the correct source, which is Xin 2018.  As you know, this is a seminal resource for qualitative analytic methods.  We removed "grounded theory" as we agree it does not apply here.

12g) What were the “more meaningful parts” that were “chunked?” (I don’t know what a “talk turn” is and couldn’t find that term anywhere in a Google search. A talk turn seems like something one might look for in discourse analysis but in this context, I couldn't speculate. 

A turn of talk is simply a way to chunk data for analysis.  We reworded it to read "sentences.

13.  Below are some smaller and editorial notes:

  • Line 29 – Omit “overly.” - Done
  • Line 151 – This is a long, difficult to understand sentence. - Revised
  • Why weren’t teachers compensated for their time?  -IRB.
  • Line 169: I didn’t know what it meant to deliver the curriculum through role playing, rotating between roles and student roles. -I'm not sure how to make the statement clearer, honestly.  Any suggestions are welcome.
  • Line 179:  – The citation [37] does not have to do with tech-based interventions so this must have been mis-cited.  - Fixed.
  • Line 222: The lesson is really 40 minutes if taught at the highest end of the recommended time. - In those teachers' specific cases, it was.  The intended time is 35 minutes.
  • Line 245: This paragraph was hard to follow. It would be helpful to break up the text by game replay, worked example, and after game play.  -Done
  • Line 270: Was this a focus group or a semi structured group interview? It’s unusual to describe a focus group as asking “each question one at a time.”  - Respectfully, I've implemented over 100 focus groups in my career, and to my knowledge, it is certainly not uncommon to ask focus group questions one at a time.  
  • Line 305: What were the drawings?   - Schematics of interactions.
  • Line 33: How was the holistic score determined?  - Already described in the paper.
  • Line 381: Typo in heading. - Fixed.

Additional comments: 

Reviewer - Passive voice and some long sentences made it a little difficult to process in scattered places.

We reviewed and amended any sentences we saw that fit these criteria.

Reviewer - I also took note that asking teachers to commit to doing lessons 3 times a week for 35 minutes is not trivial at the elementary level. So, although 6 teachers agreed to do so, this innovation is much less feasible on a larger scale. 

Please see research from researchers such as Lynn Fuchs and Sarah Powell to see evidence that such an approach is possible.  In the US, many teachers have math blocks and supplemental space where they are directed by districts to include additional programming for students who experience math difficulty under tiered models of support.  Together the research and policy in the US suggest that such an approach is scalable, and many program have done just that using similar time/use in classrooms.  For our context and program, it is certainly an empirical questions.  Thanks again for your comments.

Round 2

Reviewer 3 Report

- This paper reads well and while I didn't compare it to the original draft directly, by my recollection, it is clear and has more examples. And the inclusion of the checklist is very helpful as well. 

- I appreciate the attention to implementation measurement and game use at the elementary level. I also appreciate the focus on UDL which, while seemingly very important, wasn't very emphasized in the piece. It seems that this is one part of the "intervention" that should have been brought out more -Although UDL is referenced, it would be helpful to educate readers a little more about the CAST framework. 
- I want to note that straying from "fidelity" happens in both directions (of detracting from the intervention and enhancing it). So, while you acknowledge that some of the teachers' changes improve the use, it should be more clear that this is also not "fidelity." I suggest that you explore the implementation research literature. There you will find some frameworks that help describe what implementation is (including factors that contribute to and inhibit implementation) and that focus more on describing actual implementation rather than "integrity" (the difference between intended and actual implementation." 

- While I don't find the findings particularly compelling because a lot is known time, teacher expectations, beliefs, etc. it is interesting that the authors deconstruct "time" into other actual reasons underlying the teachers' responses. Also, it is important to have more studies in the field that explicitly measure implementation. 

- I also encourage the authors to broaden their view of implementation measurement - this checklist is very restrictive - it's not clear the expectation was that the teacher say exactly these words (?) and if not, how close do they need to be to have "integrity?" Also, rather than focus on the words themselves when measuring implementation, consider focusing on the construct - for example, for Show Slide 7, you have a script for the teacher. Is it essential that the teacher say those words or is it sufficient if the teacher uses different words but is still embodying the construct which I might call "teacher facilitation of students supporting arguments with evidence?" 

- So, all in all, I encourage you to continue focusing on implementation measurement (and focus on UDL is appreciated) and draw from literature to enhance the rigor and usability of your findings. 

A couple small things - on p. 1 - when you refer to "typical" instruction on line 27, I didn't know what you meant until I looked at the reference. So, explain what that means. 

- I'm still confused about whether observations were in person or whether "observations" are referring to listening to the audio recording. 

- It would be useful to acknowledge that the fraction test reliability is on the border of acceptable.

- page 11, line 465 - what is meant by "causal conditions?"

- I encourage you to pursue exploring an idea you raised in the conclusion - the role that asset-oriented beliefs about students plays in teachers' implementation of new materials (in this case, the game).